# Green Synthesis and Electrochemical Properties of Mono- and Dimers Derived from Phenylaminoisoquinolinequinones

**DOI:** 10.3390/molecules24234378

**Published:** 2019-11-30

**Authors:** Juana Andrea Ibacache, Jaime A. Valderrama, Judith Faúndes, Alex Danimann, Francisco J. Recio, César A. Zúñiga

**Affiliations:** 1Facultad de Química y Biología, Universidad de Santiago de Chile, Alameda 3363, casilla 40, Santiago 9170022, Chile; judith.faundes@usach.cl (J.F.); alex.danimann@usach.cl (A.D.); 2Facultad de Ciencias de la Salud, Universidad Arturo Prat, casilla 121, Iquique 1100000, Chile; 3Facultad de Química y Farmacia, Universidad Católica de Chile, casilla 306, Santiago 7820436, Chile; javier.recio@uc.cl (F.J.R.); czuniga1@uc.cl (C.A.Z.)

**Keywords:** twin drugs, heterodimers, green synthesis, amination reaction, cyclic voltammetry, half-wave potential

## Abstract

In the search for new quinoid compounds endowed with potential anticancer activity, the synthesis of novel heterodimers containing the cytotoxic 7-phenylaminoisoquinolinequinone and 2-phenylaminonaphthoquinone pharmacophores, connected through methylene and ethylene spacers, is reported. The heterodimers were prepared from their respective isoquinoline and naphthoquinones and 4,4′-diaminodiphenyl alkenes. The access to the target heterodimers and their corresponding monomers was performed both through oxidative amination reactions assisted by ultrasound and CeCl_3_·7H_2_O catalysis “in water”. This eco-friendly procedure was successfully extended to the one-pot synthesis of homodimers derived from the 7-phenylaminoisoquinolinequinone pharmacophore. The electrochemical properties of the monomers and dimers were determined by cyclic and square wave voltammetry. The number of electrons transferred during the oxidation process, associated to the redox potential E^I_1/2_^, was determined by controlled potential coulometry.

## 1. Introduction

Quinones are ubiquitous in nature and comprise one of the largest classes of anticancer agents [1,2,3]. Among the broad variety of drugs used clinically in the therapy of solid cancers, mitomycin, mitoxantrone, and saintopin contain the common quinone nucleus into their active pharmacophores. The most remarkable characteristics of these quinoid drugs are their abilities to act as DNA intercalators, reductive alkylators of biomolecules, and/or generators of reactive oxygen species (ROS) such as hydroxyl radical, hydrogen peroxide, superoxide anion, and singlet oxygen [4], which can damage tumor cells [5,6,7,8,9,10,11,12] via oxidative stress [5,13,14]. It is worth to note that in spite of the broad range of effects of quinoid compounds on grown inhibition of diverse cancer cells, the major limitations, in terms of their use as cancer drugs, are their side effects [15].

A common aminoquinoid unit appeared as a key structural scaffold in diverse natural occurring cytotoxic compounds, such as smenospongine [16,17], streptonigrin [18,19,20,21,22,23], mansouramicyn C [24,25] and, synthetic cytotoxic 1,4-benzoquinones, 1,4-naphthoquinones [26,27], and heterocyclic analogs [28,29,30,31,32].

A number of synthetic aminoisoquinolinequinones and polycyclic analogs have been the subject of study for many years due to their in vitro cytotoxic activities on several cancer cell lines [33,34,35,36,37,38,39]. A well-established procedure to prepare alkyl- and arylaminoquinones is based on the oxidative coupling reaction of quinones with alkyl- and arylamines [40,41,42,43,44]. It is widely accepted that the oxidative coupling reaction involves a Michael addition of the nitrogen nucleophiles to the quinones, followed by oxidation of the hydroquinone intermediates [44].

In Figure 1, representative synthetic phenylaminoquinones endowed with in vitro cytotoxic properties against a number of human cancer cell lines are depicted [26,30,31,32]. The common feature of these aminoquinones to target cancer cells has been attributed, in part, to their abilities to produce oxidative stress via generation of reactive oxygen species (ROS) [45,46].

The biological activity of quinones is often related to their electrochemical behavior [47,48,49,50,51,52]. The ability of the quinone nucleus to accept one or two electrons to give the corresponding semiquinone radical anion (Q^•−^) or hydroquinone dianion (Q^2−^) species is believed to induce formation of reactive oxygen species (ROS), responsible for the oxidative stress in cells [11,47,53].

Recently, we successfully applied the twin-drug approach [54] to improve the antiproliferative activity and selectivity of the phenylaminoisoquinolinequinone pharmacophore [29,55]. The designed homodimers were prepared through a one-pot procedure from phenylaminoisoquinolinequinones and 4,4′-diaminodiphenylmethane (DDM). It was also reported the selective access to monomers, which are involved in the formation of homodimers [54]. The following Scheme 1 exemplifies the selective access to the corresponding monomer or homodimer employing suitable molar ratio between the quinone and diamine precursors.

The selective access to the monomers and the cytotoxic levels of the monomers and homodimers [54] encourage us to extend our studies to the synthesis of heterodimers constituted by anilinoisoquinolinequinone and 2-anilinonaphthoquinone pharmacophores connected through methylene spacers [26]. Bearing in mind that monomers and dimers exhibit one and two electro-active quinone nucleus, respectively, we were also interested to get insight into their electrochemical properties. Herein we report results on the eco-friendly access to monomers, heterodimers, and homodimers derived from isoquinolinequinones, naphthoquinones, and 4,4′-diamino diphenyl alkanes. The new synthesized monomers and dimers were subject to electrochemical studies by cyclic voltammetry, square wave voltammetry, and controlled potential coulometry.

## 2. Results

### 2.1. Chemistry

The strategy to construct the heterodimers, where the phenylamino groups of the selected quinoid pharmacophores are connected through methylene spacers, is based on oxidative monoamination reactions of the parent isoquinolinquinones **1** and **2** with the 4,4′-diamino diphenyl alkanes **3** and **4** followed by amination reactions of the resulting monomers with naphthoquinones **8** and **9** (Figure 2).

The access to the designed heterodimers **10**, **11**, and **12** was planned through the aminoisoquinolinequinone monomers **5**, **6**, and **7** resulting from the reaction between the quinones **1**/**2** and symmetrical diamines **3**/**4**. The synthetic approach to heterodimer **10** was firstly examined from isoquinolinequinone **1**, naphthoquinone **8**, and diamine **3**. The required monomer precursor **5** was synthesized in 74% yield, according to our previously reported procedure [54], by reaction of the quinone **1** with diamine **3** in a 1:2 mole ratio, catalytic amounts of CeCl_3_·7H_2_O in ethanol at room temperature [29]. Further reaction assays of **5** with naphthoquinone **9** in a 1:2 ratio under the above-mentioned conditions, performed at room temperature and, in refluxing ethanol, produced heterodimer **10** albeit in low yields (22% and 28%, respectively). The heterodimer **10** was isolated as a purple solid, m.p. 186 °C (d). The IR spectrum reveals the presence of N–H and C=O bands at v/cm^−1^: 3337, 1720, 1666, and 1673, respectively. The ^1^H-NMR spectrum shows the signals of two vinylic protons at δ 7.54 and 7.67 ppm and the amino proton signals at δ 7.54 and 7.67 ppm. In the aromatic proton region, the proton signals of the naphthoquinone fragment, at δ: 7.67, 7.76 and 8.11, were observed and those of the protons of the phenyl groups of the linker that appeared as a multiplet at δ 7.21 ppm. The methylene protons of the spacer are observed at 4.00 ppm. The ^13^C-NMR spectrum displays signals of three carbonyl groups at δ: 181.5, 181.1, 180.4, and the mass spectrum shows the molecular ion [M+] peak at *m*/*z* 598.1971.

Interestingly, when a water suspension containing compounds **5** and **9** (1:2 mole ratio) and CeCl_3_·7H_2_O were ultrasound-irradiated [56] for 6 h, heterodimer **10** was generated and isolated in 88% yield (Scheme 2). Based on this excellent outcome, we decided to extend this green procedure to the synthesis of monomers **5**–**7** and heterodimers **11** and **12**. The reactions that were conducted under irradiation period of 1.5–7 h, produced the respective monomers **5**–**7** in excellent yields (94–98%), and compounds **11** and **12** in moderate yield (48% and 50%) (Scheme 2). The lower yields formation of heterodimers **11** and **12** compared to that of dimer **10** could be attributed to steric hindrance interactions involved in the addition of the nitrogen nucleophiles **11** and **12** across the disubstituted electrophilic quinone double bond of **9**.

The one-pot access to homodimers **13**–**15** from their respective isoquinolinquinones **1** and **2** and their symmetrical diamines **3** and **4** were also carried out under ultrasound irradiation, catalyzed with CeCl_3_·7H_2_O in water. Under these conditions, the respective homodimers **13**–**15** were isolated in good yields (65–75%) (Scheme 2).

The structures of the new compounds **6**, **7**, **10**–**12**, **14**, and **15** were established by infrared spectroscopy (IR), ^1^H- and ^13^C-nuclear magnetic resonance (NMR), bidimensional nuclear magnetic resonance (2D-NMR), and high-resolution mass spectroscopy (HRMS).

### 2.2. Electrochemistry

The monomers **5**–**7** and dimers **10**–**12**, **14**, and **15** were evaluated for their half-wave potentials (E^I_1/2_^ and E^II_1/2_^). For all the compounds, the first faradaic process (E^I_1/2_^) presents a reversible behavior, and the second process is quasi-reversible. In Table 1, the electrochemical parameters determined in this study are summarized. The net charge consumption was used to calculate the number of electrons for each compound. In the monomers (**5**–**7**), the values of charge are close to 10 mC, which indicate a one-electron transfer during the faradaic process. However, for the heterodimers and homodimers it is close to 20 mC under the studied conditions, which implies 2-electrons. These transferred electron values were corroborated by the following equation: |Epc−(Epc2)|=90.6n for an ideal reversible peak (E^I_1/2_^), where E_Pc_ is the cathodic peak potential and E_pc_/2 is potential at half-width of the cathodic peak for all the studied molecules [57,58,59].

The first potential for compounds **5**–**7** (via one-electron), and, for dimers **10**, **11**, **12**, **14**–**15** (vía two-electrons) produce the corresponding semiquinone radical anion(s) [6,13,14]. The first and second reduction potential of monomer **5** correspond to the monoelectronic transfer processes: E^I^: Q + e^−^ = semiquinone radical anion (Q^•−^) and E^II^: Q^•−^ + e^−^ = quinone dianion (Q^2−^) [60,61]. It is important to note that in the case of dimers **10**, **11**, **12**, **14**, and **15**, a broad faradaic process associated to the overlap between E^I_1/2_^ and E^II_1/2_^ is detected due to the conjugate nature of the benzene ring groups together with the fast-kinetic reaction of the carbonyl groups [62,63].

Figure 3 shows the voltammograms for monomer **5**–**7** and homo- and heterodimer **10**–**12**, **14**–**15** (Appendix A, Cyclic voltammetry and square wave voltammetry (SWV)).

The electrochemical parameters presented in Table 1 indicate that monomers **5**–**7** exhibited a relative closeness in the values of the potential E^I_1/2_^ and E^II_1/2_^ and widening of the faradic signal associated with both processes. These facts could be attributed to the electronic nature of the substituents of the compounds [46]. In the case of heterodimers **10** and its chlorine analog **11**, the shift to more positive values of E^I_1/2_^ and E^II_1/2_^ of the later with respect to **10** could be attributed to the electron-withdrawing effect of the chlorine atom. Similarly, the shift to more positive values of E^I_1/2_^ of monomer **6** compared to **7**, and homodimer **14** compared to **15** could be explained by the stronger electron-withdrawing ability of the methoxycarbonyl group in **14** than the acetyl group in **15**. These results agree with precedents on the influence of electron-withdrawing substituents in aminoquinones groups, on their oxidant properties [64,65,66,67].

The data analysis led us also to found differences between the potential E^I_1/2_^ and E^II_1/2_^ of dimers and monomers containing the 4,4′-diaminodiphenylalkane fragment in terms of the nature of the alkane spacers. The comparison of the monomers **5** and **6**, which differ in the chain length of the alkane spacers, shows that the formal potentials of the ethylene-containing monomer **6** appeared at more positive values than its methylene-containing analog 5. Additional examples are required to establish the influence of the chain length on the potential E^I_1/2_^ and E^II_1/2_^.

## 3. Materials and Methods

### 3.1. General

All solvents, reagents, and precursors such as quinones **8** and **9** were purchased from different companies such as Aldrich (St. Louis, MO, USA) and Merck (Darmstadt, Germany) and were used as supplied. Melting points were determined on a Stuart Scientific SMP3 (Bibby Sterilin Ltd., Staffordshire, UK) apparatus and are uncorrected. The IR spectra were recorded on an FT IR Bruker spectrophotometer; (model Vector 22 Bruker, Rheinstetten, Germany), using KBr disks, and the wave numbers are given in cm^−1^. ^1^H- and ^13^C-NMR spectra were recorded on a Bruker Avance-400 instrument (Bruker, Ettlingen, Germany) in CDCl_3_ at 400 and 100 MHz, respectively. Chemical shifts are expressed in ppm downfield relative to tetramethylsilane and the coupling constants (*J*) are reported in hertz. Data for ^1^H-NMR spectra are reported as follows: s = singlet, br s = broad singlet, d = doublet, m = multiplet, and the coupling constants (*J*) in Hz. Bidimensional NMR techniques were used for signal assignments. HRMS-ESI were carried out on a Thermo Scientific Exactive Plus Orbitrap spectrometer (Thermo Fisher, Bremen, Germany) with a constant nebulizer temperature of 250 °C. The experiments were performed in positive ion mode, with a scan range of *m*/*z* 100–300. All fragment ions were assigned by accurate mass measurements at high resolution (resolving power: 140,000 FWHM). The samples were infused directly into the electrospray ionization source (ESI) using a syringe pump at flow rates of 5 μL min^−1^. Silica gel Merck 60 (70–230 mesh, from Merck, Darmstadt, Germany) was used for preparative column chromatography and TLC aluminum foil 60F254 for analytical thin-layer chromatography (TLC). Isoquinolinequinones **1**–**2** were prepared by previously reported procedures [40].

The ultrasound-promoted reactions were carried out in standard oven-dried glassware in a Branson sonicator cup horn working at 19.7–20.0 kHz (75 W).

### 3.2. Chemistry

#### 3.2.1. Preparation of Monoamination Compounds **5**–**7**. General Procedure

Suspensions of quinones **1**–**2** (1 mmol) and corresponding diamine (2 equiv), CeCl_3_·7H_2_O (5 mmol %), and water (20 mL) were left with ultrasonic irradiation after completion of the reaction as indicated by TLC. The reaction mixture was partitioned between chloroform and water, the organic extract was washed with water (2 × 15 mL), dried over Na_2_SO_4_, and evaporated under reduced pressure. The residue was column chromatographed over silica gel (95:5 CH_2_Cl_2_/EtOAc) to yield the corresponding pure monoamination compounds **5**–**7**.

#### 3.2.2. Preparation of Heterodimers **10**–**12**. General Procedure

Suspensions of compounds **5**–**7** (2 mmol), naphthoquinone **8**/**9** (1 mmol), CeCl_3_·7H_2_O (5 mmol %), and water (20 mL) were left with stirring under ultrasonic irradiation after completion of the reaction as indicated by TLC. The reaction mixture was partitioned between chloroform and water, the organic extract was washed with water (2 × 15 mL), dried over Na_2_SO_4_, and evaporated under reduced pressure. The residue was column chromatographed over silica gel (95:5 CH_2_Cl_2_/EtOAc) to yield the corresponding pure heterodimers **10**–**12**.

#### 3.2.3. Preparation of Homodimers **13**–**15**. General Procedure

Suspensions of quinones **1/2** (4 mmol) and corresponding diamine **3/4** (1 equiv), CeCl_3_·7H_2_O (5 mmol %), and water (20 mL) were left with stirring under ultrasonic irradiation after completion of the reaction. The reaction mixture was partitioned between chloroform and water, the organic extract was washed with water (2 × 15 mL), dried over Na_2_SO_4_, and evaporated under reduced pressure. The residue was column chromatographed over silica gel (95:5 CH_2_Cl_2_/EtOAc) to yield the corresponding pure homodimers **13**–**15**.

*Methyl-7-(4-(4-aminobenzyl)phenyl)amino)-1,3-dimethyl-5,8-dioxo-5,8-dihydroisoquinoline-4-carboxylate* (**5**). m.p. 149–150 °C; IR (KBr) v/cm^−1^: 3423 (N-H), 3305, and 3251 (NH_2_), 1734 (C=O ester), 1668 (C=O quinone). ^1^H-NMR (400 MHz, CDCl_3_) δ 2.61 (s, 3H, 3-Me), 2.99 (s, 3H, 1-Me), 3.60 (s, 2H, NH_2_), 3.88 (s, 2H, CH_2_), 4.00 (s, 3H, CO_2_Me), 6.30 (s, 1H, 6-H), 6,62 (dd, *J* = 8.3 Hz, 12.8 Hz, 2H), 6.96 (t, *J* = 6.8 Hz, 2H), 7.14 (d, *J* = 8.3 Hz, 2H), 7.22 (d, *J* = 8.3 Hz, 2H), 7.68 (s, 1H, N-H). ^13^C-NMR (100 MHz, CDCl_3_) δ 182.1, 181.7, 169.6, 161.6, 161.3, 146.0, 145.1, 142.7, 140.8, 138.3, 130.9, 130.5, 125.5, 123.4, 120.3, 115.8, 115.7, 102.5, 53.4, 40.9, 26.5, 23.3. HRMS [M + H]^+^: calculated for C_26_H_23_N_3_O_4_: 442.1762; found: 442.1761.

*Methyl 7-(4-(4-aminophenethyl)phenylamino)-1,3-dimethyl-5,8-dioxo-5,8-dihydroisoquinoline-4 carboxylate* (**6**). m.p. 212–213 °C; IR (KBr) v/cm^−1^: 3465 (NH), 3368, and 3312 (NH_2_) 1723 (C=O ester), 1665 (C=O quinone). ^1^H-NMR (400 MHz, CDCl_3_) δ 2.64 (s, 3H, 3-Me), 2.87 (m, 4H, CH_2_CH_2_), 3.01 (s, 3H, 1-Me), 3.67 (s, 2H, NH_2_), 4.03 (s, 3H, CO_2_Me), 6.32 (s, 1H, 6-H), 6.64 (d, *J* = 8.3 Hz, 2H), 6.96 (d, *J* = 8.3 Hz, 2H), 7.18 (dd, *J* = 23.1, 8.4 Hz, 4H), 7.28 (s, 1H, NH). ^13^C-NMR (100 MHz, CDCl_3_) δ 181.7, 181.4, 169.2, 161.2, 161.0, 145.6, 144.4, 140.5, 137.8, 134.5, 131.3, 129.9, 129.3, 125.1, 122.9, 119.9, 115.3, 102.1, 53.0, 37.7, 36.9, 26.1, 22.9. HRMS [M + H]^+^: calculated for C_27_H_25_N_3_O_4_: 456.19181; found: 456.1913.

*4-acetyl-7-(4-(4-aminophenethyl)phenylamino)-1,3-dimethylisoquinoline-5,8-dione* (**7**). m.p. 224–225 °C; IR (KBr) v/cm^−1^: 3339 (NH), 3224, and 3183 (NH_2_), 1671 (C=O acetyl) and 1612 (C=O quinone), ^1^H-NMR (400 MHz, CDCl_3_) δ 2.53 (s, 3H, COMe), 2.56 (s, 3H, 3-Me), 2.85 (m, 4H, CH_2_CH_2_), 2.98 (s, 3H, 1-Me), 3.60 (s, 2H, NH_2_), 6.28 (s, 1H, 6-H), 6.62 (d, *J* = 8.3 Hz, 2H), 6.93 (d, *J* = 8.2 Hz, 2H), 7.16 (dd, *J* = 23.8, 8.4 Hz, 4H), 7.72 (s, 1H, NH). ^13^C-NMR (100 MHz, CDCl_3_) δ 22.9, 25.9, 31.1, 36.9, 37.7, 101.7, 115.3, 120.0, 123.0, 129.3, 129.9, 131.2, 133.5, 134.4, 137.9, 140.6, 144.5, 145.9, 159.8, 160.4, 181.7, 182.3, 203.8. HRMS [M + H]^+^: calcd for C_27_H_25_N_3_O_3_: 440.19689; found:440.1966.

*Methyl 7-(4-(4-(1,4-dioxo-1,4-dihydronaphthalene-2-ylamino)benzyl)phenylamino)-1,3-dimethyl-5,8-dioxo-5,8-dihydroisoquinoline-4-carboxylate* (**10**). m.p. 186 °C; IR (KBr) v/cm^−1^: 3337 (NH), 1720 (C=O ester), 1673 and 1666 (C=O quinone). ^1^H-NMR (400 MHz, CDCl_3_) δ 2.61 (s, 3H, 3-Me), 2.99 (s, 3H, 1-Me), 4.00 (s, 5H, CO_2_Me, CH_2_), 6.32 (s, 1H, CH), 6.38 (s, 1H, CH), 7.21 (m, 8H, arom), 7.54 (s, 1H, NH), 7.67 (m, 3H, NH, CH), 7.76 (t, *J* = 8.3 Hz, 1H, CH), 8.11 (t, *J* = 6.7 Hz, 1H, CH). ^13^C-NMR (100 MHz, CDCl_3_) δ 181.5, 181.1, 180.4, 177.0, 173.2, 169.3, 161.7, 162.1, 144.8, 141.3, 139.2, 138.8, 137.1, 135.5, 134.8, 133.3, 132.9, 131.3, 130.7, 130.4, 129.0, 127.7, 127.0, 124.1, 123.4, 102.5, 67.5, 53.3, 39.1, 32.7, 22.1. HRMS [M + H]^+^: calculated for C_36_H_27_N_3_O_6_: 598.19729; found: 598.1971.

*Methyl 7-(4-(4-(3-chloro-1,4-dioxo-1,4-dihydronaphthalene-2-ylamino)benzyl)phenylamino)-1,3-dimethyl-5,8-dioxo-5,8-dihydroisoquinoline-4-carboxylate* (**11**). m.p. 268–270 °C; IR (KBr) v/cm^−1^: 3323 (NH), 1719 (C=O ester), 1681 and 1678(C=O quinone), 721 (C-Cl). ^1^H-NMR (400 MHz, CDCl_3_) δ 2.61 (s, 3H, 3-Me), 2.99 (s, 3H, 1-Me), 4.00 (s, 3H, CO_2_Me), 4.01 (m, 2H, CH_2_), 6.32 (s, 1H, 6-H), 7.03 (d, *J* = 8.3 Hz, 2H), 7.17 (m, 4H, arom), 7.24 (m, 2H, arom), 7.69 (m, 3H, NH, CH), 7.76 (m, 1H, CH), 8.12 (d, *J* = 7.5 Hz, 1H, CH), 8.19 (d, *J* = 7.6 Hz, 1H, CH). ^13^C-NMR (100 MHz, CDCl3) δ 181.8, 181.5, 180.7, 177.6, 173.7, 169.3, 161.4, 161.1, 145.7, 141.7, 139.3, 138.2, 137.9, 135.9, 135.2, 133.1, 132.8, 131.0, 130.4, 130.0, 129.0, 127.3, 127.2, 124.7, 123.3, 102.4, 66.9, 53.2, 38.9, 32.1, 22,8. HRMS [M + H]^+^: calculated for C_36_H_26_ClN_3_O_6_: 632.15831; found:632.1582.

*Methyl 7-(4-(4-(3-chloro-1,4-dioxo-1,4-dihydronaphthalen-2-ylamino)phenethyl)phenylamino)-1,3-dimethyl-5,8-dioxo-5,8-dihydroisoquinoline-4-carboxylate* (**12**). m.p. 263–265 °C; IR (KBr) v/cm^−1^: 3307 (NH), 1722 (C=O ester), 1679, and 1675 (C=O quinone), 720 (C-Cl). ^1^H-NMR (400 MHz, CDCl_3_) δ 2.61 (s, 3H, 3-Me), 2.95 (s, 4H, CH_2_CH_2_), 2.99 (s, 3H, 1-Me), 4.00 (s, 3H, CO_2_Me), 6.29 (s, 1H, 6-H), 7.05 (dd, *J* = 41.2, 8 Hz, 4H, arom), 7.16 (q, *J* = 8.5 Hz, 4H, arom), 7.69 (m, 3H, CH, NH), 7.76 (t, *J* = 7.5 Hz, 1H, CH), 8.11 (d, *J* = 7.6 Hz, 1H, CH), 8.19 (d, *J* = 7.6 Hz, 1H, CH). ^13^C-NMR (100 MHz, CDCl_3_) δ 181.6, 181.4, 180.6, 177.5, 169.2, 161.2, 160.9, 145.6, 141.2, 139.7, 138.9, 137.8, 135.5, 135.0, 134.8, 132.9, 132.7, 129.9, 129.8, 128.5, 127.1, 126.9, 125.1, 124.5, 123.0, 119.9, 117.2, 114.4, 102.2, 23.0, 26.1, 29.7, 37.2. HRMS [M + H]^+^: calculated for C_37_H_28_ClN_3_O_6_: 646.17396; found: 646.1796.

*Dimethyl-7,7′-(4,4′-methylenebis(4,1-phenylene)bis(azanediyl)bis(1,3-dimethyl-5,8-dioxo-5,8- dihydroisoquinoline-4-carboxylate)* (**13**). m.p. 199–200 °C; IR (KBr): ν/cm^−1^: 3446 (NH), 1736 (C=O ester), 1652 and 1647 (C=O quinone). ^1^H-NMR (CDCl_3_) δ 2.64 (s, 6H, 3-Me), 3.02 (s, 6H, 1-Me), 4.03 (s, 8H, CH_2_ and CO_2_Me), 6.34 (s, 1H, 6-H), 7.26 (m, 8H, arom.), 7.73 (s, 2H, NH). ^13^C-NMR (100 MHz, CDCl_3_) δ 181.6, 181.4, 169.1, 161.3, 160.9, 145.5, 138.8, 137.8, 135.2, 130.2, 125.1, 123.2, 119.9, 102.3, 53.0, 40.8, 26.1, 22.93. HRMS [M + H]^+^: calculated for C_39_H_32_N_4_O_8_: 685.2293; found: 685.2208.

*Dimethyl7,7*′*-(4,4*′*-(ethane-1,2-diyl)bis(4,1-phenylene))bis(azanediyl)bis(1,3-dimethyl-5,8-dioxo-5,8-dihydroisoquinoline-4-carboxilate)* (**14**). m.p. 290–293 °C; IR (KBr) v/cm^−1^: 3315 (NH), 1730 (C=O ester), 1650 and 1649 (C=O quinone). ^1^H-NMR (400 MHz, CDCl_3_) δ 2.53 (s, 6H, 3-Me), 2.56 (s, 6H, 1-Me), 2.96 (s, 4H, CH_2_CH_2_), 2,99 (s, 6H, CO_2_Me), 6.28 (s, 2H, 6-H), 7.18 (q, *J* = 8.5 Hz, 8H, arom), 7.73 (s, 2H, NH). ^13^C-NMR (100 MHz, CDCl_3_) δ 203.8, 182.7, 182.1, 160.8, 160.2, 146.4, 140.0, 138.2, 135.3, 133.9, 130.3, 123.6, 120.4, 102.3, 37.4, 31.3, 26.2, 23.2. HRMS [M + H]^+^: calculated for C_40_H_34_N_4_O_8_: 699.24496; found: 699.2449.

*7,7*′*-(4,4*′*-(ethane-1,2-diyl)bis(4,1-phenylene))bis(azanediyl)bis(4-acetyl-1,3-dimethylisoquinoline-5,8-dione)* (**15**). m.p. 245–246 °C; IR (KBr) v/cm^−1^ 3163 (NH), 1693 (C=O acetyl). ^1^H-NMR (400 MHz, CDCl_3_) δ 2.52 (s, 6H, COMe), 2.56 (s, 6H, 3-Me), 2.96 (s, 4H, CH_2_CH_2_), 2.99 (s, 6H, 1-Me), 6.27 (s, 2H, 6-H), 7.18 (q, *J* = 8.6 Hz, 8H, arom), 7.74 (s, 2H, NH). ^13^C-NMR (100 MHz, CDCl_3_) δ 203.8, 182.3, 181.6, 160.5, 159.8, 145.9, 139.6, 137.8, 134.6, 133.3, 130.0, 123.2, 101.8, 60.4, 37.1, 31.1, 26.0, 23.0. HRMS [M + H]^+^: calculated for C_40_H_34_N_4_O_6_: 667.25513; found: 667.2570.

### 3.3. Electrochemical Measurement

The electrochemical measurements were performed in an electrochemical three electrodes cell. Calomel saturated electrode (SCE_sat_.) and platinum wire were used as a reference (implementing a Luggin capillary system) and as a counter electrode, respectively. Glassy carbon electrode was used as a working electrode (area: 0.196 cm^2^, Pine Instrument). The measurements were performed using a bipotentiostat (CH Instrument, CH1720E) in acetonitrile containing 0.1 M tetrabutylammonium perchlorate (TBAP) at room temperature. Before the measurements, the solution was deoxygenated using N_2_ as purging gas for 15 min.

The half-wave potential (E_1/2_) of the quinone compounds were characterized by cyclic voltammetry in a potential range from −1.9 or −1.5 to 0.5 V at a scan rate of 0.1 V s^−1^. The E_1/2_ was calculated as the average between the anodic and cathodic peak ((E_pa_+E_pc_)/2) [59]. In addition, square wave voltammetry (SWV) was carried out (from −1.5 to −0.5 V vs SCE) to corroborate the half-wave potential, using acetonitrile containing 0.1 M tetrabutylammonium perchlorate (TBAP) at room temperature and purging with N_2_ gas for 15 min (Appendix A, Cyclic voltammetry and square wave voltammetry (SWV)).

The number of electrons transferred in the faradaic process (E_pa_ or E_pc_) was determined by coulometric measurements. The tests were performed at a fixed potential 0.1 V higher than the highest anodic peak determined by cyclic voltammetry for two hours. The number of electrons was calculated considering the total charge (Q_net_) and using the Faraday equation that relates the charge to each mole of quinone studied. (Q_net_ = nFz,) [59] where, n = number of moles of the compound, F = Faraday constant (96.487 C mol^−1^) and z = number of transferred electrons [59]. The concentration of the compounds was 1 × 10^−5^ M in a total volume of 10 mL.

## 4. Conclusions

In conclusion, we have synthesized a series of novel heterodimers containing the cytotoxic 7-phenylaminoisoquinolinequinone and 2-phenylaminonaphthoquinone pharmacophores connected through methylene and ethylene spacers. The access to the target heterodimers and their corresponding monomers was performed both through oxidative amination reactions assisted by ultrasound and CeCl_3_·7H_2_O catalysis “in water”. This eco-friendly procedure was successfully extended to the one-pot synthesis of homodimers derived of the 7-phenylaminoisoquinolinequinone pharmacophore. For the mono and dimeric compounds, it was determined that the corresponding first potentials (E^I_1/2_^) are reversible while the second potentials (E^II_1/2_^) are quasi reversible (∆E^II_1/2_^). Furthermore, it was also established that during the oxidation process associated with the potential E^I_1/2_^, the net charge consumption for the monomers is close to 10 mC, while for the heterodimers and homodimers is nearly 20 mC. These facts indicate that in the case of homo- and heterodimers two semiquinone anion radical species are simultaneously generated in the same molecule at similar formal E^I_1/2_^ potentials (via two-electron).

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
