# Peer review of "Green Synthesis and Electrochemical Properties of Mono- and Dimers Derived from Phenylaminoisoquinolinequinones"

_molecules, 2019, doi:10.3390/molecules24234378_

Round 1

Reviewer 1 Report

Paper Green Synthesis and Electrochemical Properties of  Mono- and Dimers Derived from

 Phenylaminoisoquinolinequinones is an interesting paper concerning preparation and electrochemical properties of new quinoid derivatives. The article is well organized and the results seem interesting, I suggest some revisions before publication. In particular in the electrochemical section, I think that performing also DPV experiments can help to better understand the reduction processes. Authors claim that for compounds 5-7 there are two one-electron processes, but looking at the figure seems that the signal of first process is higher in current intensity respect to second one. DPV experiments are useful to analyze all the processes separately. Also, for compounds 10-12 and 14-15 a shoulder in CV peaks is observable, and these experiments can clarify the number of separate reductions.

Some minor point: in the experimental section commercial source or synthesis reference should be added for all the starting compounds, in particular naphthoquinones 8-9 are not indicated.

Line 108: there are two “The” at the beginning

Line 109-110 check the correspondence in verb conjugation, for example process is singular and the verb should be presents

Author Response

RESPONSE TO REVIEWER 1

The article is well organized and the results seem interesting, I suggest some revisions before publication. In particular

In the electrochemical section, I think that performing also DPV experiments can help to better understand the reduction processes. Authors claim that for compounds 5-7 there are two one-electron processes, but looking at the figure seems that the signal of first process is higher in current intensity respect to second one. DPV experiments are useful to analyze all the processes separately. Also, for compounds 10-12 and 14-15 a shoulder in CV peaks is observable, and these experiments can clarify the number of separate reductions.

Answer: We thank the reviewer for the interesting comment about the ratio between the anodic peak currents of the monomers 5-7. In this study, we also perform square wave voltammetry with no conclusive results about the number of electrons transferred during the faradaic process in monomers. Indeed, the ratio of the anodic currents and the ratio of the cathodic currents are different. The ratios between the anodic peaks are close to 2.6, and the ratios between the cathodic currents are around 1.4 electrons. During the study, we address the subject performing coulometry tests and by the study of the reversibility of the process. Both experiments conclude that there is one electron transfer for each faradic process in monomers.

In the case of the shoulders detected during the faradaic process in dimers, we associated their presence with the interaction of benzene groups with quinone groups in these molecules. This interaction induces an overlap of the faradaic processes and could also induces shoulders in the broad faradic process detected by voltammetry. We appreciate the recommendation of the use of DVP for future works.  

Some minor point: in the experimental section commercial source or synthesis reference should be added for all the starting compounds, in particular naphthoquinones 8-9 are not indicated.

Answer: In the following sentence was included the commercial source of quinonoid precursors 8 and 9: All solvents, reagents and precursors such as quinones 8 and 9 were purchased from different companies such as Aldrich (St. Louis, MO, USA) and Merck (Darmstadt, Germany) and were used as supplied.  

Line 108: there are two “The” at the beginning

Answer: It was corrected

Line 109-110 check the correspondence in verb conjugation, for example process is singular and the verb should be presents

Answer: Your right, the sentence was corrected as follow: The monomers 5-7 and dimers 10-12, 14, 15 were evaluated for their half-wave potentials (EI½ and EII½). For all the compounds, the first faradaic process (EI½) presents a reversible behavior, and the second process is quasi-reversible.

Reviewer 2 Report

This manuscript by  Ibacache, Valderrama and collaborators describes the synthesis and electrochemical properties of mono- and dimers derived from phenylaminoisoquinolinequinones. Using oxidative amination reactions assisted by ultrasound and cerium-based catalysis in water, new quinoidal compounds are produced in good yields. Finally, electrochemical studies for the respective compounds were performed showing the Redox aspects of the new compounds, intrinsically related to the potential biological activity of the new derivatives. I consider the manuscript appropriate for publication in Molecules after the insertion of minor corrections described below:

- Introduction: The authors reported the use of quinoidal derivatives as antitumor compounds. Some references related to quinoidal compounds with potential antitumor activity could be cited in the manuscript. See the following references as suggestion:

- N. Kongkathip et al.; Bioorg. Med. Chem. 11 (2003) 3179-3191; - E.L. Bonifazi et al.; Bioorg. Med. Chem. 18 (2010) 2621-2630; - P. R. Duchowicz et al.; Eur. J. Med. Chem. 77 (2014) 176-184;- T. V. Baiju et al.; Eur. J. Med. Chem. 151 (2018) 686.

- The insertion of electrochemical data would be important to corroborate the ability of the respective compounds in generate ROS. The generation of ROS and the electrochemical behavior are intrinsically related to the capacity of the quinoidal system in act as antitumor compounds. Important references related to electrochemistry studies of quinones should be cited: E.A. Hillard et al., Chem. Commun. 23 (2008) 2612; J. Org. Chem. 79 (2014) 5201; Org. Biomol. Chem. 6 (2008) 3414.

- Chemistry. I suggest the insertion of more details in the Chemistry section, such as: a) Describes about the NMR assignments. I would suggest indicating the assignment in each signal, also in the coupling constants. Choose a compound and show this data as an example. b) Authors could clarify the reason for the preparation of target compounds by the route described in the manuscript. Previously synthesis described in the literature for such class of compounds. Obviously, the authors cited a reference of the group on the synthesis of a compound related to the new derivatives described here, but the discussion based on the literature could be more detailed, if it is possible. I would like to see some preparation for understanding of the readers. Mechanism discussion and etc.

Author Response

RESPONSE TO THE REVIEWER 2

This manuscript by Ibacache, Valderrama and collaborators describes the synthesis and electrochemical properties of mono- and dimers derived from phenylaminoisoquinolinequinones. Using oxidative amination reactions assisted by ultrasound and cerium-based catalysis in water, new quinoidal compounds are produced in good yields. Finally, electrochemical studies for the respective compounds were performed showing the Redox aspects of the new compounds, intrinsically related to the potential biological activity of the new derivatives. I consider the manuscript appropriate for publication in Molecules after the insertion of minor corrections described below:

Introduction: The authors reported the use of quinoidal derivatives as antitumor compounds. Some references related to quinoidal compounds with potential antitumor activity could be cited in the manuscript. See the following references as suggestion: N. Kongkathip et al.; Bioorg. Med. Chem. 11 (2003) 3179-3191; - E.L. Bonifazi et al.; Bioorg. Med. Chem. 18 (2010) 2621-2630; - P. R. Duchowicz et al.; Eur. J. Med. Chem. 77 (2014) 176-184;- T. V. Baiju et al.; Eur. J. Med. Chem. 151 (2018) 686.

Answer: We thank your valuable comments and according to your suggestions, the references [33-36], related to quinoidal compounds with antitumor activity were included in the manuscript.

The insertion of electrochemical data would be important to corroborate the ability of the respective compounds in generate ROS. The generation of ROS and the electrochemical behavior are intrinsically related to the capacity of the quinoidal system in act as antitumor compounds. Important references related to electrochemistry studies of quinones should be cited: E.A. Hillard et al., Chem. Commun. 23 (2008) 2612; J. Org. Chem. 79 (2014) 5201; Org. Biomol. Chem. 6 (2008) 3414.

Answer: Considering your suggestions on electrochemistry of quinones, the above references were included in the manuscript [50-52].

Chemistry. I suggest the insertion of more details in the Chemistry section, such as: a) Describes about the NMR assignments. I would suggest indicating the assignment in each signal, also in the coupling constants. Choose a compound and show this data as an example.

Answer: According to your comments, a description of the spectral properties of heterodimer 10 was included in the manuscript.

The heterodimer 10 was isolated as purple solid, m.p. 186 ° (d). The IR spectrum reveals the presence of N–H and C=O bands at v/cm−1: 3337, 1720, 1673 and 1666. The 1H NMR spectrum shows two vinyl proton signals at δ: 6.32 and 6.38 and, the amino proton signals at δ:7.54 and 7.67 ppm. In the aromatic proton region, it was observed the pattern signals of the naphthoquinone fragment, at δ: 7.67, 7.76 and 8.11 and those of the protons of the phenyl groups of the linker at δ 7.21. The proton signals of the CH2 and CO2CH3 groups appeared overlapped at δ 4.00. The 13C NMR spectrum displays signals of three carbonic groups at δ: 180.4, 181.1 and 181.5, and the mass spectrum shows the molecular ion [M+] peak at m/z 598.1971. 

Authors could clarify the reason for the preparation of target compounds by the route described in the manuscript. Previously synthesis described in the literature for such class of compounds. Obviously, the authors cited a reference of the group on the synthesis of a compound related to the new derivatives described here, but the discussion based on the literature could be more detailed, if it is possible. I would like to see some preparation for understanding of the readers. Mechanism discussion and etc.

Answer: The synthetic strategy to prepare our target compounds was based on the oxidative amination reaction of quinones with amines. This is a general and well-known procedure for the preparation of biological active aminoquinoid compounds. The following phrase was included in the introduction section.   

A number of synthetic aminoisoquinolinequinones and polycyclic analogues have been the subject of study for many years due to their in vitro cytotoxic activities on several cancer cell lines [33–39]. A well-established procedure to prepare alkyl- and arylaminoquinones is based on the oxidative coupling reaction of quinones with alkyl- and arylamines [40–44]. It is widely accepted that the oxidative coupling reaction involves a Michael addition of the nitrogen nucleophiles to the quinones, followed by oxidation of the hydroquinone intermediates [44].
